# Comparison of the Ocular Surface Disease Index and the Symptom Assessment in Dry Eye Questionnaires for Dry Eye Symptom Assessment

**DOI:** 10.3390/life13091941

**Published:** 2023-09-21

**Authors:** Raul Martin

**Affiliations:** Instituto Universitario de Oftalmobiología Aplicada (IOBA), Universidad de Valladolid, Paseo de Belén, 17—Campus Miguel Delibes, 47011 Valladolid, Spain

**Keywords:** Ocular Surface Disease Index (OSDI), Symptom Assessment iN Dry Eye (SANDE), agreement, score normalization

## Abstract

Background: Patient-reported dry eye symptoms (DESs), assessed using the Ocular Surface Disease Index (OSDI) and the Symptom Assessment iN Dry Eye (SANDE) questionnaires, were compared in a large sample of patients. Methods: The correlation (Spearman coefficient) and agreement (Bland-Altman analysis) between the OSDI and SANDE questionnaire scores (with and without score normalization) were assessed in 1033 patients and classified according to the OSDI score as non-DES and DES in a cross-sectional analysis. Results: The normalized and non-normalized SANDE results were higher than the OSDI results in all samples (2.83 ± 12.40 (*p* = 0.063) and 2.85 ± 15.95 (*p* = 0.016), respectively) and in non-DES (*p* > 0.063) and DES (*p* < 0.001) with both OSDI cutoff values. Weak correlations were found (Spearman coefficient < 0.53; *p* < 0.001) in all cases except DES (0.12, *p* = 0.126). Weak agreement was found with a Bland-Altman analysis of the normalized and non-normalized scores of both questionnaires (mean difference from −7.67 ± 29.17 (DES patients) to −1.33 ± 8.99 (non-DES patients) without score normalization, and from −9.21 ± 26.37 (DES patients) to −0.85 ± 4.01 (non-DES) with data normalization), with a statistically significant linear relationship (R^2^ > 0.32, *p* < 0.001). The SANDE questionnaire did not yield the same patient classification as OSDI. The same operative curves (ROC) of the SANDE normalized and non-normalized scores were used to differentiate among patients with DES using OSDI < 12 (0.836 ± 0.015) or OSDI < 22 (0.880 ± 0.015) cutoff values. Conclusions: Normalized and non-normalized data collected from the SANDE questionnaire showed relevant differences from those of the OSDI, which suggests that the results of the SANDE visual analog scale-based questionnaire provide different patient classifications than the OSDI score.

## 1. Introduction

Dry eye disease (DED) is a worldwide, inflammatory, multifactorial ocular surface disease with high prevalence (affecting 5% to 50% of the population) that deteriorates the quality of daily life [1,2]. Major risk factors of DED include older age, female sex, and others such as race, video-display terminal use, cataract surgery, contact lens use, ocular surface diseases (pterygium, meibomian gland dysfunction), topical eye medication use, systemic disease, and living region (with different disease distribution in Eastern and Western countries) [3]. DED represents one of the most frequent reasons for ophthalmic consultations, but its diagnosis is a challenge [4] due to the variety of symptoms [5] and inconsistency of disease signs [2,6]. In this regard, some reports suggest that more attention to the condition from ophthalmologists would be beneficial [1,3].

For these reasons, questionnaires that record patients’ symptoms are widely used in DED diagnosis and follow-up. Such questionnaires include the Impact of Dry Eye in Everyday Life questionnaire [7], the University of North Carolina Dry Eye Management Scale [8], the Dry Eye-Related Quality-of-Life Score questionnaire [9], the Ocular Comfort Index questionnaire [10], the Standard Patient Evaluation of Eye Dryness questionnaire [11], the McMonnies Questionnaire [12,13], the dry eye questionnaire DEQ-5 [14], and others. The Ocular Surface Disease Index (OSDI) and Symptom Assessment in Dry Eye (SANDE) have been widely used for decades [15,16,17,18,19,20,21,22] because they provide reliable and valid measurements of dry eye symptoms (DES) [15,17]. The OSDI questionnaire was developed by Allergan Inc. (Irvine, CA, USA) in 2000 [15], and the current version includes a 12-item self-administered questionnaire that evaluates the frequency of symptoms over the preceding week in approximately 5 min. The OSDI score ranges between 0 and 100, where higher scores represent greater severity of symptoms, as follows: no symptoms (score ≤ 12), mild symptoms (score between 13 and 22), moderate symptoms (score between 23 and 32), and severe symptoms (score between 33 and 100) [15,23]. The OSDI score assesses the severity of symptomatology [14,24], distinguishes between patients with and without DES (with 12 or 22 as cutoff values), [2,15,16,17,25], and is commonly used in epidemiological studies [18,26]. Although the development process of the OSDI questionnaire was not reported, [16], it is one the most widely used questionnaires to discriminate between patients, with adequate psychometric properties (exploratory factor analysis [15] and Rash analysis [17]) and reliability (reproducibility) [15]. The SANDE questionnaire makes it possible to quickly and intuitively quantify both the frequency and severity of DES with just two questions using a 100-mm visual analog scale (VAS). The measurement of symptom frequency ranges from “rarely” to “all of the time”, and the measurement of symptom severity ranges from “very mild” to “very severe” [27,28]. The SANDE questionnaire has a weak correlation with clinical tests (corneal staining) but a strong correlation with treatment [28]. Additionally, a moderate correlation with the OSDI score (after score normalization) has been observed in patients with mild to severe DED [18].

Some reports have compared OSDI and SANDE results in dry eye patients [18,19] and proposed the use of the SANDE questionnaire to allow clinicians to quickly and reliably measure DES [18] to conduct the disease diagnosis (with a cutoff value of 30 [20]) and to assess the effect of different treatments [21,22]. However, there is a lack of studies comparing OSDI and SANDE results in a large sample of patients with and without DES.

This study aimed to compare assessments of DES using the OSDI and SANDE questionnaires in a large sample of patients with and without DES while also exploring the potential of the SANDE questionnaire to differentiate between patients with and without DES.

## 2. Materials and Methods

The OSDI and SANDE scores from patients who attended a routine eye exam in 12 primary eye care centers of the EMO research group in Spain in a diverse range of geographical locations (Figure 1) were compared and analyzed. All patients were evaluated in a single visit, and both questionnaires were conducted via interviews by the investigators; the same application protocol was followed in all centers. A comprehensive eye examination was performed, including visual acuity measurement (Snellen optotypes), manifest refraction (with phoropter or trial frame following four steps: initial sphere check, cylinder axis refinement, cylinder power refinement, and second sphere check), and anterior ocular surface assessment, including fluorescein dye observation under a slit lamp using cobalt blue illumination. Additionally, an extensive clinical history to detect risk factors associated with DED was conducted, and patients with any disorder affecting tear secretion (such as hyperthyroidism, rheumatism, lupus, any autoimmune disease, previous diagnosis of DED, cicatricial conjunctivitis, pterygium, eyelid trichiasis, or others) [29], a history of any eye medication use in the last 3 months to treat any eye condition (such as glaucoma), active anterior eye inflammation (such as blepharitis, conjunctivitis, keratitis, scleritis, or uveitis), the use of contact lenses in the last 3 months, or any previous ocular trauma or surgery in the last 6 months were excluded. All researchers were trained in the study protocol to minimize the impact of the inter-practitioner variability in the study procedures.

The institutional review board of the Human Sciences Ethics Committee of Valladolid Area-Este Clinic Hospital (Castilla y Leon public health system-SACYL) approved the study protocol (PI-201606), and the study was conducted according to the tenets of the Helsinki Declaration. All patients were informed about the nature of the study and its consequences prior to obtaining their written informed consent to participate.

DES were evaluated with the OSDI and SANDE questionnaires. The OSDI questionnaire [15] is structured into three main domains as follows: ocular symptoms (5 questions), vision-related daily function (4 questions), and environmental triggers (3 questions). Patients answered the twelve questions with a Likert-type scale from 0 to 4 as follows: 0 indicates never; 1, some of the time; 2, half of the time; 3, most of the time; and 4, all of the time. To calculate the total OSDI score, the sum of scores multiplied by 100 was divided by the total number of questions answered multiplied by 4 [23]. Additionally, to provide comparable results with previous reports, patients were classified according to the most widely recommended OSDI cutoff values in two categories: non-DES (OSDI ≤ 12 and OSDI ≤ 22) and DES (OSDI > 13 and OSDI > 23) [2]. The SANDE questionnaire comprises 2 questions as follows: “How often do your eyes feel dry and/or irritated?” and “How severe do you feel your symptoms of dryness and/or irritation are?” on a 100-mm horizontal VAS. For patients marked on both lines, the severity and frequency of their symptoms and distance were measured in millimeters. The final SANDE score was calculated by multiplying the frequency score by the severity score and obtaining the square root [27,28].

For statistical analysis, SPSS software version 27.0 (SPSS, Inc., Chicago, IL, USA) was used. Normal distribution of the variables was assessed with the Kolmogorov-Smirnov test (*p* < 0.05 indicated that the data were normally distributed). Means, standard deviations (SDs), and percentages were used to describe the data where appropriate. All statistical analyses were considered significant at *p* < 0.05 [30].

To compare the OSDI and SANDE scores, a previous report [18] recommended normalizing the scale (applying the algebraic method in the norm of a vector), because the two questionnaires do not measure symptoms in the same way. Therefore, a statistical comparison (correlation and agreement) was conducted with normalized and non-normalized scores to compare the effect of normalization.

The correlations between data obtained from both questionnaires were assessed using linear regression analysis and the Spearman coefficient of correlation. The differences between the scores of both questionnaires were analyzed using a paired Wilcoxon signed-rank test. The graphs of the differences between the pairs of measurements obtained by each questionnaire divided by the average of the means of each pair of readings were plotted, and the limits of agreement (LoA) were calculated (mean of the difference ± 1.96 × SD), as suggested by Bland and Altman [31]. Linear regression analysis was used to assess the effect of the overall magnitude of the mean distance on the differences between the results of both questionnaires, and the R^2^ correlation coefficient was calculated. Exact 95% confidence intervals (CI) for the repeatability limits of agreement were also calculated and plotted [32]. Both analyses (correlation and difference) were conducted for all samples and in non-DES and DES patients classified with OSDI scores (with 12 and 22 cutoff values) [2].

The differences in SANDE score according to OSDI classification groups were assessed with Kruskal-Wallis ANOVA with Bonferroni correction [33] (pairwise comparison between OSDI groups of non-DES, mild, moderate, and severe DES). Additionally, differences between non-DES and DES classified with OSDI cutoff values of 12 or 22 were assessed with the Mann-Whitney U test. Finally, to explore the use of the SANDE questionnaire to classify DES patients (as the OSDI questionnaire classifies with cutoff values of 12 or 22 [2]), a receiver operating characteristic (ROC) curve was plotted, and the area under the curve was calculated.

## 3. Results

A total of 1033 patients, with an average age of 52.6 ± 14.8 years (ranging from 18 to 97 years), a mean spherical equivalent (sphere + ½ cylinder) of −0.07 ± 2.23 D (ranging from +7.75 to −15.00 D), and best spectacle-corrected visual acuity of 20/25 or better, completed the OSDI and SANDE questionnaires. A total of 405 (39.2%) were male and 628 (60.8%) were female.

On the basis of the scores generated by the OSDI questionnaire, of the 1033 patients evaluated, 742 (71.8%) were classified as non-DES (OSDI score ≤ 12) and 291 (28.2%) were classified as DES (OSDI score > 13) (131 (6.6%) mild, 76 (7.4%) moderate and 84 (8.1%) severe DES). Using the OSDI cutoff value of 22, a total of 873 (84.5%) patients were classified as non-DES (OSDI score ≤ 22), and 160 (15.4%) were classified as DES (OSDI score > 22).

The scores of the OSDI and SANDE questionnaires with and without normalization are summarized in Table 1. Without normalization, the OSDI and SANDE questionnaires showed statistically significant differences (*p* = 0.016) for all samples and DES patients (classified with both OSDI −12 or 22 cutoff values-) (*p* < 0.01); however, non-DES patients showed no statistically significant differences between the two questionnaire results (*p* > 0.684). After score normalization, the overall sample did not show statistically significant differences (*p* = 0.063) between the OSDI and SANDE scores; however, as with non-normalized data, DES patients showed significantly different results (*p* ≤ 0.001), and non-DES patients provided similar results (*p* > 0.063). The Spearman correlation coefficient results revealed a significant correlation (*p* < 0.001) between OSDI and SANDE scores, except for those of moderate and severe DES patients (OSDI >23) (R = 0.12; *p* = 0.126), with and without score normalization (Figure 2).

Bland-Altman analysis for clinical agreement between non-normalized OSDI and SANDE scores (Figure 3) revealed a clinical difference (bias) from −7.67 (DES patients with OSDI > 23) to −1.33 (non-DES patients with OSDI ≤ 12). A similar trend was found when comparing normalized data (Figure 4), where agreement ranged from −9.21 (DES patients with OSDI > 23) to −0.85 (non-DES patients with OSDI ≤ 12). However, all differences (with and without normalization) showed significant linear correlation with the mean value (R^2^ > 0.31; *p* < 0.001) (Figure 3 and Figure 4).

The SANDE score showed a significantly different value (*p* < 0.001) in patients classified with the OSDI score as being without DES (5.60 ± 9.51; 95% CI from 4.91 to 6.28) or with mild (22.70 ± 21.80; 95% CI from 18.30 to 25.83), moderate (40.98 ± 26.67; 95% CI from 34.88 to 47.07), or severe (46.19 ± 29.52; 95% CI from 39.78 to 52.59) symptoms (Figure 5—left). However, a pairwise comparison showed statistically non-significant differences between SANDE scores in moderate and severe DES patients (*p* = 0.656) classified with OSDI score. Statistically significant differences were found in the SANDE score in patients with and without DES classified with OSDI cutoff values of 12 and 22 (Figure 5—right).

ROC analysis (Figure 6) was used to assess the SANDE scores to differentiate between patients with and without DES; it showed a similar area under the curve with both OSDI cutoff values (0.836 ± 0.015; 95% CI from 0.807 to 0.866 with OSDI cutoff values of 12 and 0.880 ± 0.015; 95% CI from 0.851 to 0.908 with OSDI cutoff values of 22) and limited values of sensitivity and specificity to distinguish between patients with and without DES. Normalized and non-normalized SANDE scores yielded the same area under the curve.

## 4. Discussion

Several questionnaires have been proposed to explore DES in clinical and research practice [7,8,9,10,11], but OSDI and SANDE have been widely used for decades [15,16,17,18,19,20,21,22]. The OSDI questionnaire, in conjunction with DED, has been recommended as the primary questionnaire for diagnosing DED by TFOS DEWIS [14]. Although most comparisons between the OSDI and SANDE questionnaires have assessed patients with dry eyes [18,19], both questionnaires have been proposed to differentiate between patients with and without DES [15,16,17,18,20,25,27,28] and to evaluate the disease progression and treatment effect [15,21,22,23,34]. For this reason, we compared OSDI and SANDE scores in a large sample of patients with and without DES to clarify if both questionnaires could provide interchangeable results to distinguish between patients with different dryness severity symptoms in eye care practice.

Although the non-normalized SANDE score was statistically significant between patients with and without DES according to the OSDI classification (Figure 4), both questionnaires showed significantly different values with and without score normalization (Table 1) in patients with DES. Additionally, a weak correlation (Spearman coefficient lower than 0.53) in all compared samples and DES subgroups (Figure 2) was found between both questionnaire results. Previous reports found correlation coefficients of 0.64 [18] and 0.67 [19] between both questionnaires in dry eye patients, i.e., slightly higher than our results (correlation coefficients of 0.40 and 0.12, Figure 2, using OSDI cutoffs of 12 and 22, respectively).

The differences between both questionnaires suggest that normalized and non-normalized SANDE scores are slightly higher than OSDI results (Table 1) in all of the compared samples and subgroups. Chen et al. found [35] significantly higher SANDE scores in a sample of young women, mostly without dry eyes, and Kheirkhah et al. [22] also found higher SANDE (67.5 ± 17.8) than OSDI (47.9 ± 23.2) scores in a sample of patients with meibomian gland dysfunction. In contrast, other reports have described lower SANDE scores than OSDI scores in dry eye patients [18,19,21].

After score normalization, a small difference (ranging from −7.67 to −1.33) compared with previous reports of 16 units [18] was found, albeit with similar differences between OSDI and SANDE scores with and without score normalization (Figure 3 and Figure 4). This suggests that score normalization does not provide a great advantage in data comparisons. In contrast to previous reports, a linear tendency (Figure 3 and Figure 4) between the difference and mean value of the OSDI and SANDE questionnaires was found; as such, we do not recommend their interchangeable use.

Although both questionnaires showed small differences and the SANDE score was significantly different between patients with and without DES (classified with both OSDI cutoff values), the ROC analysis suggested a limited value of sensitivity and/or specificity (Figure 6) with the SANDE questionnaire compared to OSDI when classifying DES patients. Wang et al. [20]. found a similar area under the curve (higher than 0.80) and higher sensitivity (86%) and specificity (94%) combining the SANDE cutoff ≥ 30, noninvasive tear film break-up time < 10 s, tear film lipid layer grade ≤ 3, and tear meniscus height < 0.2 mm. However, no cross validation of these results has been reported.

The differences found in this study compared with previous reports could be related to several factors. For example, previous reports compared OSDI and SANDE results in dry eye patients [18,19] who may have already been familiar with these questionnaires. However, in our study, a large sample of patients (with and without DES) who attended a routine eye exam in an eye care center were asked to complete the OSDI and SANDE questionnaires; it was therefore to be expected that most of them had not previously completed this type of questionnaire. The effect of patients or subjects who answer questionnaires could be related to the psychometric properties of the OSDI and SANDE questionnaires. A recent report [16] reviewed the properties of questionnaires designed to explore DES according to the COnsensus-based Standards for the selection of health Measurement Instruments (COSMIN) Risk of Bias checklist and found that the psychometric properties (content validity, measurement error, and structural validity) of these questionnaires were not clearly assessed or described. The differences found in our study could be explained by the lack of previous validation of the questionnaires compared in this study. Our findings highlight the necessity of a revalidation of these questionnaires based on current methodological standards and recommendations [16,36] to provide better tools to assess DES in clinical and research practice and facilitate comparisons of future results.

Additionally, another limitation of this study is related to the use of the OSDI questionnaire to classify patients with and without DES, which may result in the misclassification of DED (which requires an assessment of clinical signs and tear osmolality). However, both questionnaires (OSDI and SANDE) are widely used to explore dryness symptomatology, especially OSDI, which has been validated to distinguish between healthy and DED patients [2,15,16,17,25,36], and the results of this comparison could be of great utility in eye care practice and research.

## 5. Conclusions

In conclusion, this study, based on a large population assessed with the OSDI and SANDE questionnaires, shows that the results of both questionnaires cannot be used interchangeably in eye care practice. The normalization of SANDE questionnaire scores may not be necessary to compare results, because normalized and non-normalized data collected from the SANDE questionnaire showed relevant differences from those of the OSDI, which suggests that the results of the SANDE visual analog scale-based questionnaire provide different patient classifications (in terms of distinguishing between patients with and without DES) than the OSDI score. Recommendations in future reports to improve the psychometric validation of DES questionnaires and to standardize the procedure to describe patient symptomatology for dry eyes could help provide useful research results and improve the management of DES patients.

## Figures and Tables

**Figure 1 life-13-01941-f001:**
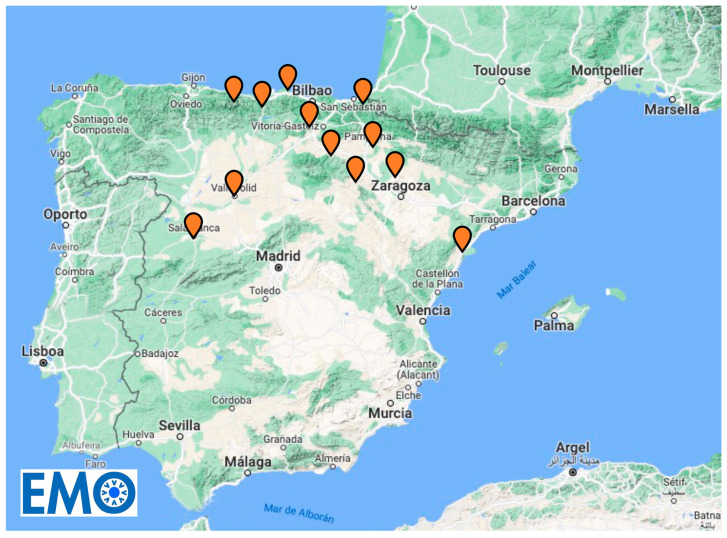
Representation of the geographical locations of the EMO (Optometry Multicentre Studies) group centers.

**Figure 2 life-13-01941-f002:**
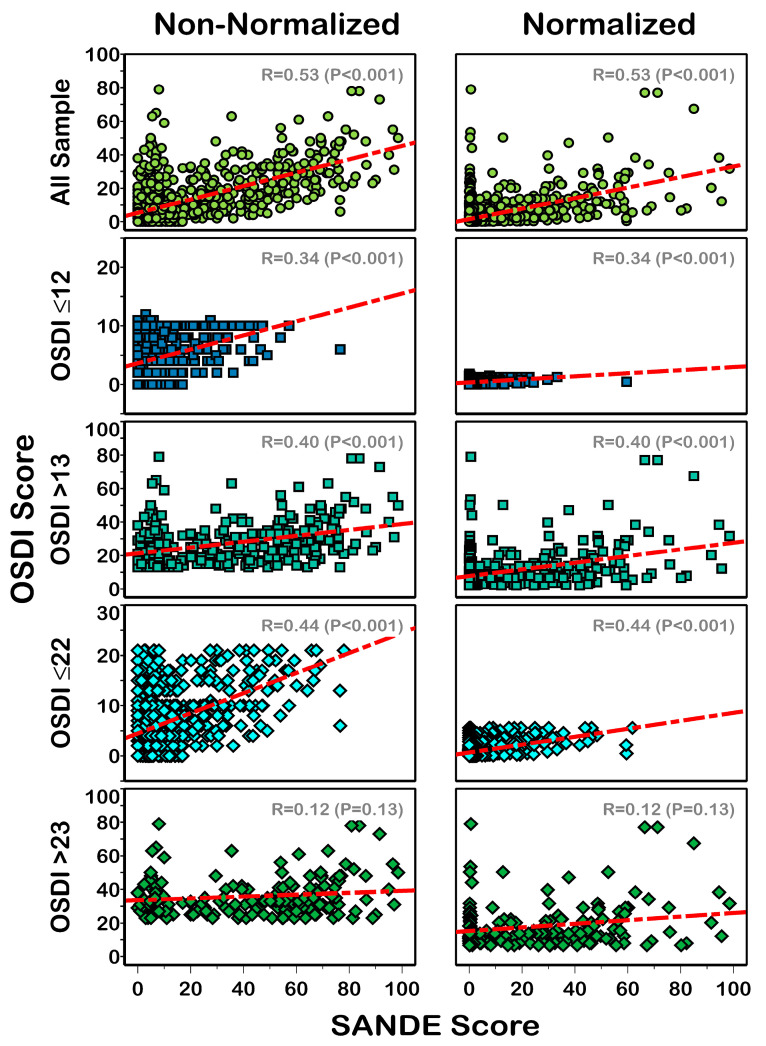
Scatterplots showing the correlation between OSDI and normalized (**right**) and non-normalized (**left**) SANDE scores from patients with and without DES. R value = Spearman correlation coefficient. Red dotted lines represent correlation between data.

**Figure 3 life-13-01941-f003:**
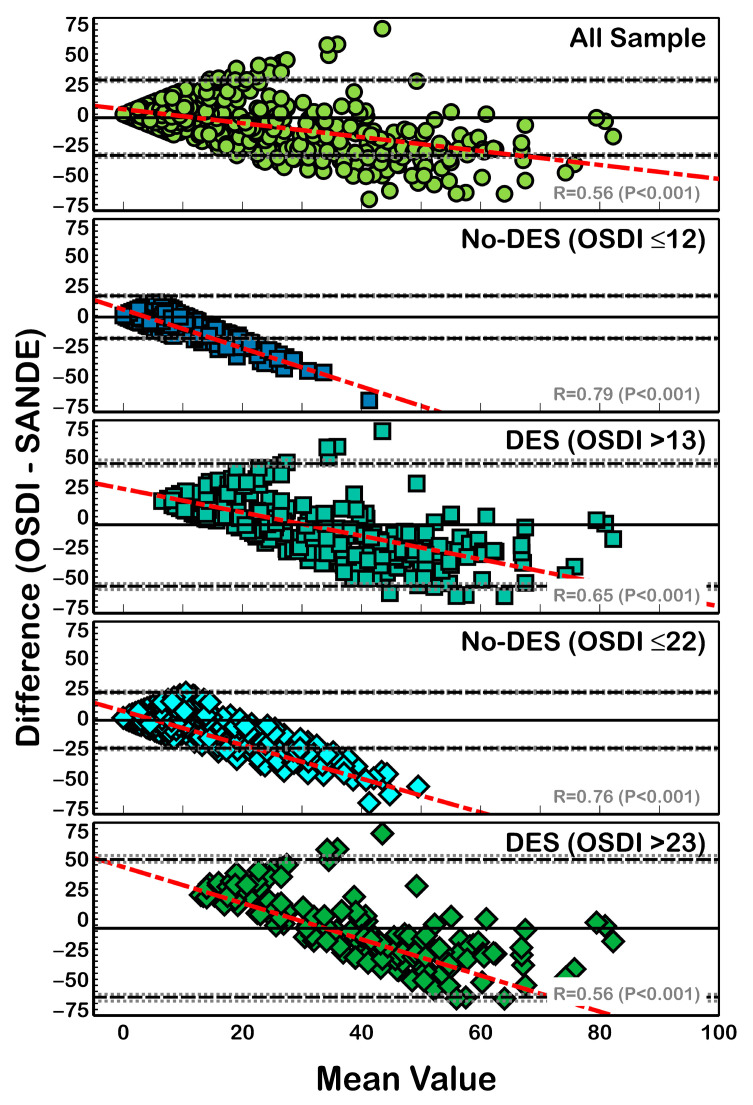
Bland-Altman plot for the differences between normalized OSDI and SANDE scores. Black solid lines show the mean difference, and dotted lines show the LoA. Gray dotted lines show the 95% CI of LoA. A linear regression between the difference and mean is also shown. All sample scores showed a mean difference of −2.85 ± 15.92 units, with LoA (95% CI) that ranged from 28.35 (27.05 to 30.10) to −34.05 (−32.75 to −35.80) units, and the correlation coefficient (R^2^) was 0.32 (*p* < 0.001). With an OSDI cutoff value of 12, non-DES patients showed a mean difference of −1.33 ± 8.99 units, with LoA (95% CI) that ranged from 16.30 (15.56 to 17.29) to −18.96 (−18.22 to −19.95) units and R^2^ = 0.62 (*p* < 0.001), and DES patients showed a mean difference of −6.74 ± 25.96 unit, with LoA (95% CI) that ranged from 44.15 (42.02 to 47.00) to −57.63 (−55.50 to −60.48) units and R^2^ = 0.43 (*p* < 0.001). Finally, with an OSDI cutoff value of 22, non-DES patients showed a mean difference of −1.97 ± 11.82 units, with LoA (95% CI) that ranged from 21.19 (20.23 to 22.49) to −25.13 (−24.17 to −26.43) units and R^2^ = 0.58 (*p* < 0.001). DES patients showed a mean difference of −7.67 ± 29.17 units, with LoA (95% CI) that ranged from 49.51 (47.12 to 52.72) to −64.85 (−62.46 to −68.06) units and R^2^ = 0.49 (*p* < 0.001). Red dotted lines represent correlation between data.

**Figure 4 life-13-01941-f004:**
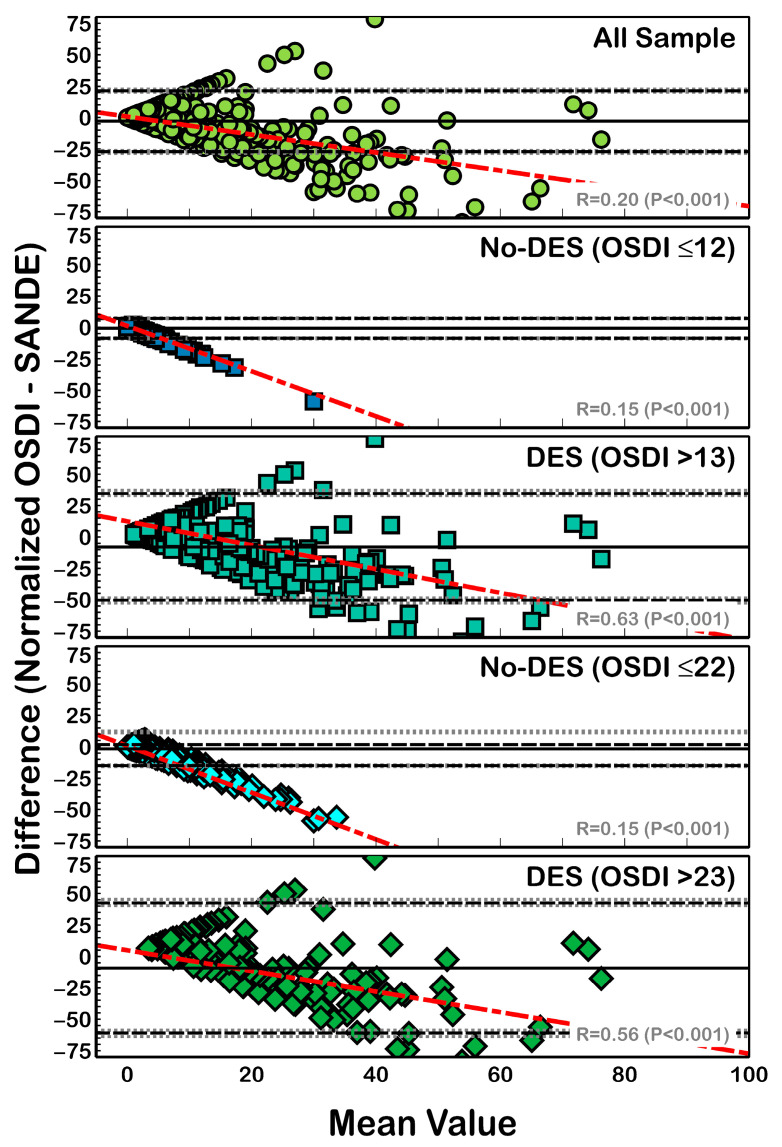
Bland-Altman plot for the differences between normalized OSDI and SANDE scores. Black solid lines show the mean difference, and dotted lines show the LoA. Gray dotted lines show the 95% CI of LoA. The linear regression between the difference and mean is also shown. All sample scores showed a mean difference of −2.83 ± 12.4 units, with LoA (95% CI) that ranged from 21.47 (20.45 to 22.83) to −27.13 (−26.11 to −28.49) units, and the correlation coefficient (R^2^) was 0.32 (*p* < 0.001). With an OSDI cutoff value of 12, non-DES patients showed a mean difference of −0.85 ± 4.01 units, with LoA (95% CI) that ranged from 7.02 (6.69 to 7.46) to −8.72 (−8.39 to −9.16) units, and R^2^ = 0.61 (*p* < 0.001), and DES patients showed a mean difference of −7.86 ± 21.69 units, with LoA (95% CI) that ranged from 34.64 (32.87 to 37.03) to −50.36 (−48.59 to −52.75) units and R^2^ = 0.13 (*p* < 0.001). Finally, with an OSDI cutoff value of 22, non-DES patients showed a mean difference of −1.66 ± 6.8 units, with LoA (95% CI) that ranged from 11.66 (11.1 to 12.41) to −14.98 (−14.42 to −15.73) units and R^2^ = 0.21 (*p* < 0.001). DES patients showed a mean difference of −9.21 ± 26.37 units, with LoA (95% CI) that ranged from 42.48 (40.32 to 45.38) to −60.9 (−58.74 to −63.8) units and R^2^ = 0.04 (*p* = 0.012). Red dotted lines represent correlation between data.

**Figure 5 life-13-01941-f005:**
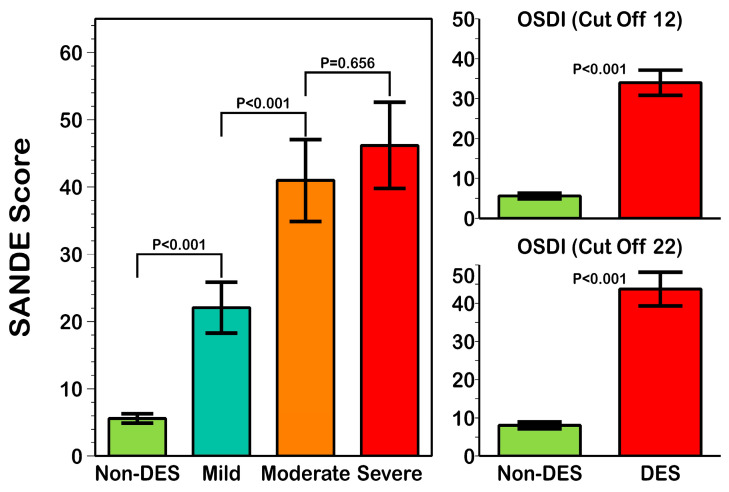
Summary of the SANDE score differences in each group of patients classified with the OSDI score. (**Left**) Differences in each OSDI group of patients, where the Kruskal-Wallis ANOVA *p* value with Bonferroni correction is provided to show OSDI group differences. (**Right**) Differences between patients with and without DES using OSDI cutoff values of 12 (**top**) and 22 (**bottom**); the Mann-Whitney U test *p* value is provided to show differences with both OSDI cutoff values.

**Figure 6 life-13-01941-f006:**
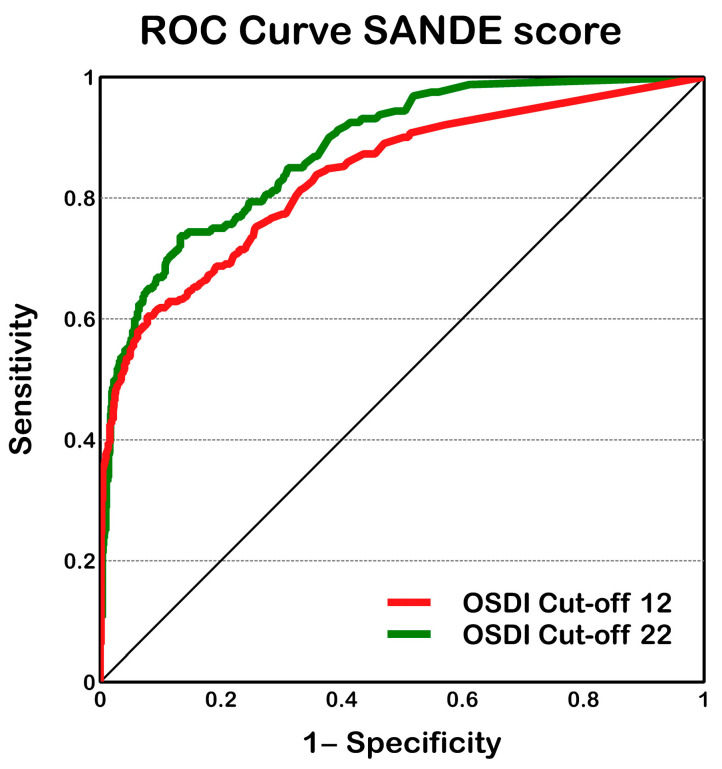
SANDE ROC curve. ROC curve to differentiate, with normalized and non-normalized SANDE scores, between patients with and without DES, classified with OSDI cutoff values of 12 (red line) or 22 (green line).

**Table 1 life-13-01941-t001:** Comparison of OSDI and SANDE scores with and without normalization.

	Non-Normalized Scores Mean ± SD (95% CI)				Normalized Scores Mean ± SD (95% CI)			
	OSDI	SANDE	*p* Value *	Rho Spearman	OSDI	SANDE	*p* Value *	Rho Spearman
All Samples (n = 1033)	10.74 ± 12.84 (9.95 to 11.52)	13.59 ± 21.06 (12.30 to 14.87)	0.016	0.53 (*p* < 0.001)	3.54 ± 8.35 (3.03 to 4.05)	6.37 ± 15.03 (5.45 to 7.29)	0.063	0.53 (*p* < 0.001)
OSDI Cutoff = 12								
Non-DES (n = 742)	4.27 ± 3.44 (4.02 to 4.52)	5.60 ± 9.51 (4.91 to 6.28)	0.684	0.34 (*p* < 0.001)	0.38 ± 0.43 (0.35 to 0.41)	1.23 ± 4.1 (0.94 to 1.53)	0.063	0.34 (*p* < 0.001)
DES (n = 291)	27.23 ± 13.27 (25.70 to 28.76)	33.97 ± 27.68 (30.77 to 37.16)	<0.001	0.40 (*p* < 0.001)	11.61 ± 12.52 (10.16 to 13.05)	19.47 ± 22.83 (16.83 to 22.1)	<0.001	0.40 (*p* < 0.001)
** *p* Value	<0.001	<0.001	-	-	<0.001	<0.001	-	-
OSDI Cutoff = 22								
Non-DES (n = 873)	6.10 ± 5.49 (5.73 to 6.46)	8.07 ± 13.50 (7.17 to 8.96)	0.840	0.44 (*p* < 0.001)	0.85 ± 1.27 (0.77 to 0.94)	2.51 ± 7.28 (2.03 to 2.99)	0.998	0.44 (*p* < 0.001)
DES (n = 160)	36.04 ± 11.91 (34.18 to 37.90)	43.71 ± 28.23 (39.30 to 48.12)	<0.01	0.12 (*p* = 0.126)	18.23 ± 13.66 (16.1 to 20.36)	27.44 ± 25.43 (23.47 to 31.41)	0.001	0.12 (*p* = 0.126)
** *p* Value	<0.001	<0.001	-	-	<0.001	<0.001	-	-

* *p* value = Wilcoxon signed-rank test. ** *p* value = U Mann–Whitney.

## Data Availability

The data presented in this study are available on request from the corresponding author.

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
