# Peer review of "Comparison of the Ocular Surface Disease Index and the Symptom Assessment in Dry Eye Questionnaires for Dry Eye Symptom Assessment"

_life, 2023, doi:10.3390/life13091941_

Round 1

Reviewer 1 Report

Certainly, there are some questions, the authors could consider and focus on them again sturdily:

The authors are required to clarify these items in the manuscript Questionnaire Selection Rationale, Questionnaire Validity and Reliability, Questionnaire Comparison, and Cross-Cultural Adaptation.

Was the sample size of 1033 participants large enough to provide statistically meaningful results?

Were the participants recruited from a diverse range of demographics and geographical locations, or could there be potential biases in participant selection?

How were the OSDI and SANDE questionnaires validated for assessing dry eye symptoms?

Were the OSDI and SANDE questionnaires equally effective in capturing the nuances of dry eye symptoms, or could one questionnaire be more sensitive than the other?

While weak correlations were observed between the OSDI and SANDE scores, could other factors, such as individual variability in interpreting symptoms, impact these correlations?

In the Bland-Altman analysis, were the calculated mean differences clinically significant, or were they within an acceptable range considering the nature of dry eye symptoms? More focusing.

How was the process of normalization carried out for the SANDE scores, and could this process introduce any unintended biases?

How do the differences in patient classification between OSDI and SANDE questionnaires impact clinical decision-making and treatment strategies for dry eye patients?

Could the differences in patient classification be due to the wording, structure, or visual analog scale used in the questionnaires?

Were there any existing studies that have compared OSDI and SANDE questionnaires, and how do the findings of this study align with or contradict those previous findings?

Could the differences observed in this study be attributed to the specificity of the symptoms captured by each questionnaire?

Could a longitudinal study design provide more insights into the consistency and reliability of the observed differences between OSDI and SANDE scores over time?

Were the symptoms of dry eye stable among participants, or could they have varied during the study period, affecting the questionnaire results?

How well do the ROC curves perform in differentiating patients with dry eye using the different cutoff values for OSDI scores? Were there any false positives or false negatives?

Given the discrepancies in patient classification between OSDI and SANDE scores, what recommendations would the authors provide to clinicians using these questionnaires for diagnosing and treating dry eye? Clarify them again.

Minor editing

Author Response

Reviewer #1: Reviewer comments:

Certainly, there are some questions, the authors could consider and focus on them again sturdily:

The authors are required to clarify these items in the manuscript Questionnaire Selection Rationale, Questionnaire Validity and Reliability, Questionnaire Comparison, and Cross-Cultural Adaptation.

Comment.

We are not sure the meaning of this comment. We believe that description of these items requires an excessive large Introduction that could be not adequate, because Introduction section must provide the research rationale and objectives without extensively reviewing the literature.

Was the sample size of 1033 participants large enough to provide statistically meaningful results?

Comment.

Sample size is enough to compare both questionnaires and guarantee statistical comparison because we explore their agreement in patients without previous diagnoses of DED. Previous reports have compared both questionnaires in shorter samples that sample compared in this study and because statistical differences have been found increase sample size can not guarantee different results (or could be possible that with smaller sample differences can not be statistically significant due to a sample bias).

Were the participants recruited from a diverse range of demographics and geographical locations, or could there be potential biases in participant selection?

Comment.

In this multicentre study 12 different clinics recruited participants and are located in diverse range area of the north of Spain to avoid biases in participants selection. We have included a new figure (Figure 1) to clarify the wide range of geographical locations and we have clarified this issue in Material & Method section.

How were the OSDI and SANDE questionnaires validated for assessing dry eye symptoms?

Comment.

Yes, both questionnaires are widely used to assess dry eye symptoms as we highlight in introduction section (Lines 57-74) with references #16 (A. Recchioni et all 2021), #15 (R.M. Schiffman et all, 2000), #17 (B.E. Dougherty et all 2011), #28 (A. Gulati et all 2006) or #29 (D.A. Schaumberg et all 2007) and are validated for assessing dry eye symptoms.

Were the OSDI and SANDE questionnaires equally effective in capturing the nuances of dry eye symptoms, or could one questionnaire be more sensitive than the other?

Comment.

This is a great question, that we try to help to clarify. Previous report (Amparo et all, 2015) suggests that SANDE score could be normalized and “correlate with negligible score differences with OSDI score”. However, our results demonstrate that both questionnaires provide non-interchangeable results, and more research could be necessary to know what questionnaire could be more sensitive.

While weak correlations were observed between the OSDI and SANDE scores, could other factors, such as individual variability in interpreting symptoms, impact these correlations?

Comment.

This is an interesting comment, because same participants answered both questionnaires in same visit, it is expected that individual variability impact in results must be minimal.

In the Bland-Altman analysis, were the calculated mean differences clinically significant, or were they within an acceptable range considering the nature of dry eye symptoms? More focusing.

Comment.

This is an interesting comment. It is expected that differences could not be clinically significant, but because a statistically significant negative linear relationship exits between difference and mean value, the difference it is not a constant value. So, difference is higher (with negative value) when higher score, so this study clearly demonstrate that mean difference can not be used as “reference value” because it is not constant.

How was the process of normalization carried out for the SANDE scores, and could this process introduce any unintended biases?

Comment.

We followed previously described normalized method (Amparo et all 2015), so this process can not introduce unintended biases.

How do the differences in patient classification between OSDI and SANDE questionnaires impact clinical decision-making and treatment strategies for dry eye patients?

Comment.

As we summarized in Figure 5, SANDE score could allow a classification of patients with mild dry eye symptoms, but can not differentiate between moderate to severe dry eye symptomatology patients. So, these results suggest that patient classification could be different, and clinicians be aware of this in decision-making. But it is important to highlight that in this study non diagnosed DED patients have been included, and this could mean that in primary eye care practice both questionnaires could help to differentiate patients with dryness symptomatology (that it is different from conduct a DED diagnosis).

Could the differences in patient classification be due to the wording, structure, or visual analog scale used in the questionnaires?

Comment.

This is an interesting comment. Honestly, could be difficult (or may be impossible) to assess the impact of wording, structure, or visual analogue scale in questionnaire score. But because both are validated questionnaires it is expected a limited impact of these factors in resulted score and patient classification. So, differences in patient classification should be related with the aspects that each questionnaire assess for.

Were there any existing studies that have compared OSDI and SANDE questionnaires, and how do the findings of this study align with or contradict those previous findings?

Comment.

Yes, as we highlight in our manuscript, Amparo et all 2015 and 2018 compared both questionnaire in dry eye patients with different results than those we have found (for this reason we strongly believe that these results are useful for future clinicians). We clearly discussed these differences in our discussion section.

Could the differences observed in this study be attributed to the specificity of the symptoms captured by each questionnaire?

Comment.

Yes, it could be possible because both questionnaires explore different aspects of the patient's symptoms. The OSDI mainly evaluates the frequency of symptoms while SANDE evaluates severity. But, clinicians cand decided which questionnaire should be adequate in each case.

Could a longitudinal study design provide more insights into the consistency and reliability of the observed differences between OSDI and SANDE scores over time?

Comment.

This is an interesting comment and a great idea to future research. Longitudinal study design involving participants with and without dry eye will be of great interest to see the consistency and reliability of OSDI and/or SANDE score.

Were the symptoms of dry eye stable among participants, or could they have varied during the study period, affecting the questionnaire results?

Comment.

Reviewer’s comment is right, patients’ symptoms vary with time. For this reason, we collected both questionnaires in the same participant in the same visit, to avoid any possible impact of variability with time in our study results. As Reviewer notes in previous comment, a longitudinal prospective study could be necessary to clarify this issue.

How well do the ROC curves perform in differentiating patients with dry eye using the different cutoff values for OSDI scores? Were there any false positives or false negatives?

Comment.

To answer this interesting question could be necessary a different study design including participants with diagnosed dry eye to assess the patient classification provided with SANDE questionnaire using the cut-off values found in this study. But ROC curve included in our results suggest limited values of sensitivity and specificity to distinguish between patients with and without DES (as we noted in our manuscript).

Given the discrepancies in patient classification between OSDI and SANDE scores, what recommendations would the authors provide to clinicians using these questionnaires for diagnosing and treating dry eye? Clarify them again.

Comment.

The main recommendation that results of this study allow is that normalization of SANDE data is not necessary (as we include in study conclusions) and also that SANDE questionnaire provides different patient classification than OSDI questionnaire (also noted in study conclusions). But, each clinician must decide what questionnaire is adequate in each case according patients history, symptoms and signs and type of clinic. For example, in clinics that do not manage dry eye patients SANDE questionnaire could be (with caution) a short way to detect DES and decide to refer to a dry eye clinic, but in clinics than manage dry eye patients OSDI questionnaire could be a better option.

Reviewer 2 Report

Manuscript no. life-2595651

Manuscript title:  Comparison of the Ocular Surface Disease Index and the Symptom Assessment in Dry Eye Questionnaires for Dry Eye Symptom Assessment 

Comments:

Abstract:

SANDE is it internationally known abbreviation? I dont agree with ‘’iN’’ PART OF SANDE

Introduction:

Try to differentiate between dry eye disease and keratoconus as both are dry eye disease.

Explain factors causing dry eye disease and make a diagram of all these factors

Discussion:

First and second paragraph of discussion fit well in introduction part

Always start your discussion with your objectives and your findings

Highlighted paragraphs are like introduction part. there is no correlation between study findings and these paragraphs.

Discuss your novel findings and then relate to them with previously reported data in both agreement or disagreement manner

Conclusion:

conclusion should be brief and specific as mentioned in the abstract

Author Response

Reviewer #2: Reviewer comments

Abstract:

SANDE is it internationally known abbreviation? I dont agree with ‘’iN’’ PART OF SANDE

Comment.

SANDE is de acronym of the Symptom Assessment iN Dry Eye questionnaire. It is not an internationally abbreviation and it was described by Schaumberg et all 2007 (Reference #17).

Introduction:

Try to differentiate between dry eye disease and keratoconus as both are dry eye disease.

Comment.

We are not sure to understand this comment, keratoconus is an eye disease, but this condition is not the dry eye disease. We followed internationally accepted definition of DED (TFOS DEWS II Epidemiology Report).

Explain factors causing dry eye disease and make a diagram of all these factors

Comment.

Many factors affect to dry eye and TFOS DEWS II workshop has clearly described them. We believe that include a diagram with these factors can not provide useful information to future readers of this study, because the main objective of this work is not related with these factors, the main objective was to compare two questionnaires.

Discussion:

First and second paragraph of discussion fit well in introduction part

Comment.

First paragraph and most of the content of the second paragraph have been moved to Introduction section following Reviewer’s suggestion.

Always start your discussion with your objectives and your findings

Comment.

We included our objectives in second paragraph of Discussion section (now first paragraph lines 256-266), but we have modified last sentence of this paragraph to clarify our objective, addressing this Reviewer comment.

Highlighted paragraphs are like introduction part. there is no correlation between study findings and these paragraphs.

Comment.

All highlighted paragraphs have been revised and modified following Reviewer recommendations.

Discuss your novel findings and then relate to them with previously reported data in both agreement or disagreement manner

Comment.

We have discussed results of our study and agreement and disagreement with previous reported results. For example, Lines 268-276 discuss differences in correlation coefficients between both questionnaires, Lines 278-284 discuss differences between OSDI and SANDE scores and compare with previous reports, Lines 286-292 discuss the impact of SANDE score normalization and differences with previous reports that recommend this normalization, and finally, Lines 294-301 discuss the utility of SANDE score (with and without normalization) to classify patients with dryness symptomatology. In all discussion section we present our results and compare with previous reported.

Conclusion:

conclusion should be brief and specific as mentioned in the abstract.

Comment.

Conclusion section in main manuscript allows more detail and information than available in Abstract section. We have rewritten Conclusion to maintain specific information provided in Abstract, as Reviewer recommends.

Reviewer 3 Report

In this document, the authors aim to compare two specific questionnaires for assessing dry eye symptoms, the OSDI and SANDE, and to evaluate the diagnostic capability of SANDE. This topic is of significant interest as it necessitates the use of dependable and validated tools for diagnosis. The statistical analysis is thorough and well-executed. Below, I provide a series of comments intended to enhance the manuscript.

General Comment:

- Patient vs. Participant: The terms "patient" and "participant" are used interchangeably in the manuscript. While it is acknowledged that all individuals attended a hospital or a similar centre for evaluation, some may not exhibit symptoms and, therefore, do not fit the strict definition of "patients," as they are healthy individuals. Given the nature of the study, this reviewer recommends standardizing the terminology and employing the term "participant" consistently throughout the document.

- CI and LOA: These acronyms are utilized in the figure legends (e.g., Figure 1, Figure 2...) but are not employed or explicitly mentioned in the main text (line 122, line 125) or in the figures and tables where they are referenced (e.g., Table 1). I recommend a comprehensive review of the entire manuscript to ensure uniformity in terminology and acronym usage.

Other Comments:

- Lines 49 – 52: While it is recognized that the primary focus is on the questionnaires examined in the document, the authors should acknowledge other validated questionnaires with extensive publications, such as the McMonnies (e.g., 10.1155/2016/6210853, 10.2147/IJGM.S410790) or the DEQ-5, which is recommended alongside OSDI by TFOS for diagnosing DED (e.g., 10.1016/j.jtos.2017.05.001).

- Lines 72-75: The objective is somewhat unclear or difficult to read. It could be refined based on the following statement: "This study aimed to compare the assessment of DES using OSDI and SANDE questionnaires in a large participant population, both with and without DES, while also investigating the potential of the SANDE questionnaire to differentiate between these two groups."

- Line 79: It is understood that the same questionnaire application protocol was followed for all cases. It would be beneficial to specify whether this was self-administered or conducted via interviews by the investigators.

- Lines 79-85: I recommend adding references to this sentence (e.g., 10.1016/j.clae.2022.101800).

- Lines 117-120: I suggest including a reference here to justify the utilization of this statistical method (e.g., 10.1111/j.1475-1313.2010.00815.x).

- Lines 130-131: It is advisable to add a reference here to support the use of this statistical method (e.g., 10.1111/opo.12131).

- Line 135: "Operative curve (ROC)" should be corrected to "Receiver Operating Characteristic (ROC) curve."

- Lines 139 – 142: Descriptive statistics are provided here (spherical equivalent and visual acuity) that are not established as inclusion criteria or mentioned in the methodology. Are they truly necessary? If so, it would be helpful to briefly indicate in the methodology when and how these measurements were taken.

- Lines 143 -147: I recommend splitting this paragraph into two sentences separated by a period, as it would enhance clarity by addressing two distinct ideas: individuals with DES and those without, and among those with DES, the breakdown by severity level.

- Line 152: A p-value should be included.

- Lines 152 – 155: Each of the statements reported here should include a p-value.

- Line 171: To maintain consistency with the notation used throughout the manuscript, I recommend modifying the "R2 coefficient higher than 0.31" to "R2>0.31."

- Line 213: The acronym was previously introduced (line 135); therefore, it should be simply referenced here.

- Lines 231-233: I recommend adding that the OSDI, in conjunction with the DED, has been recommended as the primary questionnaire for diagnosing DED by TFOS DEWIS. This inclusion will underscore the relevance of the present study (e.g., 10.1016/j.jtos.2017.05.001).

- Line 239: "Dryness symptomatology" should consistently be referred to as "DES" for terminology consistency throughout the manuscript.

- Line 276: I recommend deleting this "section title" as it does not appear necessary.

The quality of English language is at an appropriate level

Author Response

Reviewer #3: Reviewer comments

In this document, the authors aim to compare two specific questionnaires for assessing dry eye symptoms, the OSDI and SANDE, and to evaluate the diagnostic capability of SANDE. This topic is of significant interest as it necessitates the use of dependable and validated tools for diagnosis. The statistical analysis is thorough and well-executed. Below, I provide a series of comments intended to enhance the manuscript.

General Comment:

- Patient vs. Participant: The terms "patient" and "participant" are used interchangeably in the manuscript. While it is acknowledged that all individuals attended a hospital or a similar centre for evaluation, some may not exhibit symptoms and, therefore, do not fit the strict definition of "patients," as they are healthy individuals. Given the nature of the study, this reviewer recommends standardizing the terminology and employing the term "participant" consistently throughout the document.

Comment.

Thanks for this comment, we have replaced participants by patient, throughout all manuscript.

- CI and LOA: These acronyms are utilized in the figure legends (e.g., Figure 1, Figure 2...) but are not employed or explicitly mentioned in the main text (line 122, line 125) or in the figures and tables where they are referenced (e.g., Table 1). I recommend a comprehensive review of the entire manuscript to ensure uniformity in terminology and acronym usage.

Comment.

We have revised and defined acronyms throughout all manuscript. CI and LOA acronyms have defined in Lines 147 (LoA) and 151 (CI). Sorry for this mistake.

Other Comments:

- Lines 49 – 52: While it is recognized that the primary focus is on the questionnaires examined in the document, the authors should acknowledge other validated questionnaires with extensive publications, such as the McMonnies (e.g., 10.1155/2016/6210853, 10.2147/IJGM.S410790) or the DEQ-5, which is recommended alongside OSDI by TFOS for diagnosing DED (e.g., 10.1016/j.jtos.2017.05.001).

Comment.

McMonnies and DEQ-5 questionnaires have been included with recommended references, following Reviewer’s recommendation.

- Lines 72-75: The objective is somewhat unclear or difficult to read. It could be refined based on the following statement: "This study aimed to compare the assessment of DES using OSDI and SANDE questionnaires in a large participant population, both with and without DES, while also investigating the potential of the SANDE questionnaire to differentiate between these two groups."

Comment.

This sentence has been replaced with Reviewer’ proposal with some little changes (for example the term participant has been replaced by patient as this Reviewer recommends in previous comment).

- Line 79: It is understood that the same questionnaire application protocol was followed for all cases. It would be beneficial to specify whether this was self-administered or conducted via interviews by the investigators.

Comment.

The procedure followed to conduct both questionnaires was clarified (conducted via interviews by the investigators), following Reviewer’s comment.

- Lines 79-85: I recommend adding references to this sentence (e.g., 10.1016/j.clae.2022.101800).

Comment.

Reference has been included following Reviewer’s recommendation.

- Lines 117-120: I suggest including a reference here to justify the utilization of this statistical method (e.g., 10.1111/j.1475-1313.2010.00815.x).

Comment.

Reference has been included following Reviewer’s recommendation.

- Lines 130-131: It is advisable to add a reference here to support the use of this statistical method (e.g., 10.1111/opo.12131).

Comment.

Reference has been included following Reviewer’s recommendation.

- Line 135: "Operative curve (ROC)" should be corrected to "Receiver Operating Characteristic (ROC) curve."

Comment.

Text has modified following reviewer recommendation.

- Lines 139 – 142: Descriptive statistics are provided here (spherical equivalent and visual acuity) that are not established as inclusion criteria or mentioned in the methodology. Are they truly necessary? If so, it would be helpful to briefly indicate in the methodology when and how these measurements were taken.

Comment.

Reviewer comment is right. Refraction and visual acuity are not inclusion criteria of this study. But this information is usually provided and help to future readers to know the clinical characteristics of the studied sample of patients. We have included a description of the procedures conducted in study visit to clarify this issue following Reviewer’s recommendation.

- Lines 143 -147: I recommend splitting this paragraph into two sentences separated by a period, as it would enhance clarity by addressing two distinct ideas: individuals with DES and those without, and among those with DES, the breakdown by severity level.

Comment.

This is an interesting comment, in this sentence we try to show the number of patients without DES (742) and with DES (291) with a description of the severity of symptoms [mild (131), moderate (76), and severe (84)]. If this information is separated in two sentences could result confuse to future readers. But following Reviewer’s recommendation we have modified this sentence to clarify.

- Line 152: A p-value should be included.

- Lines 152 – 155: Each of the statements reported here should include a p-value.

Comment (to the reviewer's two suggestions).

In this paragraph we decided to include just more relevant P value (because all details are presented in Table 1), but following Reviewer’s comment P value has been included after each of the statements reported in this paragraph (lines 179-184).

- Line 171: To maintain consistency with the notation used throughout the manuscript, I recommend modifying the "R2 coefficient higher than 0.31" to "R2>0.31."

Comment.

We have modified text following Reviewer’s recommendation.

- Line 213: The acronym was previously introduced (line 135); therefore, it should be simply referenced here.

Comment.

We have modified text following Reviewer’s recommendation.

- Lines 231-233: I recommend adding that the OSDI, in conjunction with the DED, has been recommended as the primary questionnaire for diagnosing DED by TFOS DEWIS. This inclusion will underscore the relevance of the present study (e.g., 10.1016/j.jtos.2017.05.001).

Comment.

This sentence has been included, as Reviewer recommends.

- Line 239: "Dryness symptomatology" should consistently be referred to as "DES" for terminology consistency throughout the manuscript.

Comment.

We have modified text following Reviewer’s recommendation.

- Line 276: I recommend deleting this "section title" as it does not appear necessary.

Comment.

We have deleted section title following Reviewer’s recommendation.
